# On the Importance of Feature Separability in Predicting Out-Of-Distribution Error

**Renchunzi Xie**[1]    **Hongxin Wei**[2*]    **Lei Feng**[1]    **Yuzhou Cao**[1]    **Bo An**[1]

[1] School of Computer Science and Engineering, Nanyang Technological University, Singapore
[2] Department of Statistics and Data Science, Southern University of Science and Technology, China
`{xier0002,yuzhou002}@e.ntu.edu.sg`
`weihx@sustech.edu.cn`
`lfengqaq@gmail.com`
`boan@ntu.edu.sg`

## Abstract

Estimating the generalization performance is practically challenging on out-of-distribution (OOD) data without ground-truth labels. While previous methods emphasize the connection between distribution difference and OOD accuracy, we show that a large domain gap not necessarily leads to a low test accuracy. In this paper, we investigate this problem from the perspective of feature separability empirically and theoretically. Specifically, we propose a dataset-level score based upon feature dispersion to estimate the test accuracy under distribution shift. Our method is inspired by desirable properties of features in representation learning: high inter-class dispersion and high intra-class compactness. Our analysis shows that inter-class dispersion is strongly correlated with the model accuracy, while intra-class compactness does not reflect the generalization performance on OOD data. Extensive experiments demonstrate the superiority of our method in both prediction performance and computational efficiency.

## 1   Introduction

Machine learning techniques deployed in the open world often struggle with distribution shifts, where the test data are not drawn from the training distribution. Such issues can substantially degrade the test accuracy, and the generalization performance of a trained model may vary significantly on different shifted datasets [Quinonero-Candela et al., 2008, Koh et al., 2021]. This gives rise to the importance of estimating out-of-distribution (OOD) errors for AI Safety [Deng and Zheng, 2021]. However, it is prohibitively expensive or unrealistic to collect large-scale labeled examples for each shifted testing distribution encountered in the wild. Subsequently, predicting OOD error becomes a challenging task without access to ground-truth labels.

In the literature, a popular direction in predicting OOD error is to utilize the model output on the shifted dataset [Jiang et al., 2021, Guillory et al., 2021, Garg et al., 2022], which heavily relies on the model calibration. Yet, machine learning models generally suffer from the overconfidence issue even for OOD inputs [Wei et al., 2022], leading to suboptimal performance in estimating OOD performance. Many prior works turned to measuring the distribution difference between training and OOD test set, due to the conventional wisdom that a higher distribution shift normally leads to lower OOD accuracy. AutoEval [Deng and Zheng, 2021] applies Fréchet Distance to calculate the distribution distance for model evaluation under distribution shift. A recent work [Yu et al., 2022]

---

*Corresponding Author

37th Conference on Neural Information Processing Systems (NeurIPS 2023).

introduces Projection Norm (ProjNorm) metric to measure the distribution discrepancy in network parameters. However, we find that the connection between distribution distance and generalization performance does not always hold, making these surrogate methods to be questionable. This motivates our method, which directly estimates OOD accuracy based on the feature properties of test instances.

In this work, we show that feature separability is strongly associated with test accuracy, even in the presence of distribution shifts. Theoretically, we demonstrate that the upper bound of Bayes error is negatively correlated with inter-class feature distances. To quantify the feature separability, we introduce a simple dataset-level statistic, Dispersion Score, which gauges the inter-class divergence from feature representations, i.e., outputs of feature extractor. Our method is motivated by the desirable properties of embeddings in representation learning [Bengio et al., 2013]. To achieve high accuracy, we generally desire embeddings where different classes are relatively far apart (i.e., high inter-class dispersion), and samples in each class form a compact cluster (i.e., high intra-class compactness). Surprisingly, our analysis shows that intra-class compactness does not reflect the generalization performance, while inter-class dispersion is strongly correlated with the model accuracy on OOD data.

Extensive experiments demonstrate the superiority of Dispersion Score over existing methods for estimating OOD error. First, our method dramatically outperforms existing training-free methods in evaluating model performance on OOD data. For example, our method leads to an increase of the $R^2$ from 0.847 to 0.970 on TinyImageNet-C [Hendrycks and Dietterich, 2019] – a 14.5% of relative improvement. Compared to the recent ProjNorm method [Yu et al., 2022], our method not only achieves superior performance by a meaningful margin, but also maintains huge advantages in computational efficiency and sample efficiency. For example, using CIFAR-10C dataset as OOD data, Dispersion Score achieves an $R^2$ of 0.972, outperforming that of ProjNorm (i.e., 0.947), while our approach only takes around 3% of the time consumed by ProjNorm.

Overall, using Dispersion Score achieves strong performance in OOD error estimation with high computational efficiency. Our method can be easily adopted in practice. It is straightforward to implement with deep learning models and does not require access to training data. Thus our method is compatible with modern settings where models are trained on billions of images.

We summarize our main contribution as follows:

1. We find that the correlation between distribution distance and generalization performance does not always hold, downgrading the reliability of existing distance-based methods.

2. Our study provides empirical evidence supporting a significant association between feature separability and test accuracy. Furthermore, we theoretically show that increasing the feature distance will result in a decrease in the upper bound of Bayes error.

3. We propose a simple dataset-level score that gauges the inter-class dispersion from feature representations, i.e., outputs of feature extractor. Our method does not rely on the information of training data and exhibits stronger flexibility in OOD test data.

4. We conduct extensive evaluations to show the superiority of the Dispersion Score in both prediction performance and computational efficiency. Our analysis shows that Dispersion Score is more robust to various data conditions in OOD data, such as limited sample size, class imbalance and partial label set. Besides, we show that intra-class compactness does not reflect the generalization performance under distribution shifts (SubSection 4.4).

## 2 Problem Setup and Motivation

### 2.1 Preliminaries: OOD performance estimation

**Setup** In this work, we consider multi-class classification task with $k$ classes. We denote the input space as $\mathcal{X}$ and the label space as $\mathcal{Y} = \{1, \ldots, k\}$. We assume there is a training dataset $\mathcal{D} = \{\boldsymbol{x}_i, y_i\}_{i=1}^n$, where the $n$ data points are sampled *i.i.d.* from a joint data distribution $\mathcal{P}_{\mathcal{X}\mathcal{Y}}$. During training, we learn a neural network $f : \mathcal{X} \to \mathbb{R}^k$ with trainable parameters $\theta \in \mathbb{R}^p$ on $\mathcal{D}$.

In particular, the neural network $f$ can be viewed as a combination of a feature extractor $f_g$ and a classifier $f_\omega$, where $g$ and $\omega$ denote the parameters of the corresponding parts, respectively. The feature extractor $f_g$ is a function that maps instances to features $f_g : \mathcal{X} \to \mathcal{Z}$, where $\mathcal{Z}$ denotes the

feature space. We denote by $\boldsymbol{z}_i$ the learned feature of instance $\boldsymbol{x}_i$: $\boldsymbol{z}_i = f_g(\boldsymbol{x}_i)$. The classifier $f_\omega$ is a function from the feature space $\mathcal{Z}$ to $\mathbb{R}^k$, which outputs the final predictions. A trained model can be obtained by minimizing the following expected risk:

$$\mathcal{R}_{\mathcal{L}}(f) = \mathbb{E}_{(\boldsymbol{x},y)\sim\mathcal{P}_{\mathcal{X}\mathcal{Y}}}\left[\mathcal{L}\left(f(\boldsymbol{x};\theta),y\right)\right]$$
$$= \mathbb{E}_{(\boldsymbol{x},y)\sim\mathcal{P}_{\mathcal{X}\mathcal{Y}}}\left[\mathcal{L}\left(f_\omega(f_g(\boldsymbol{x})),y\right)\right]$$

**Problem statement**  At test time, we generally expect that the test data are drawn from the same distribution as the training dataset. However, distribution shifts usually happen in reality and even simple shifts can lead to large drops in performance, which makes it unreliable in safety-critical applications. Thus, our goal is to estimate how a trained model might perform on the shifted data without labels, i.e., unlabeled out-of-distribution (OOD) data.

Assume that $\tilde{\mathcal{D}} = \{\tilde{\boldsymbol{x}}_i\}_{i=1}^m$ be the OOD test dataset and $\{\tilde{y}_i\}_{i=1}^m$ be the corresponding unobserved labels. For a certain test instance $\tilde{\boldsymbol{x}}_i$, we obtain the predicted labels of a trained model by $\tilde{y}_i' = \arg\max f(\tilde{\boldsymbol{x}}_i)$. Then the ground-truth test error on OOD data can be formally defined as:

$$\text{Err}(\tilde{\mathcal{D}}) = \frac{1}{m}\sum_{i=1}^m \mathbb{1}(\tilde{y}_i' \neq \widetilde{y}_i), \tag{1}$$

To estimate the real OOD error, the key challenge is to formulate a score $S(\tilde{\mathcal{D}})$ that is strongly correlated with the test error across diverse distribution shifts without utilization of corresponding test labels. With such scores, a simple linear regression model can be learned to estimate the test error on shifted datasets, following the commonly used scheme [Deng and Zheng, 2021, Yu et al., 2022].

While those output-based approaches suffer from the overconfidence issue [Hendrycks and Gimpel, 2016, Guillory et al., 2021], other methods primarily rely on the distribution distance between training data $\mathcal{D}$ and test data $\tilde{\mathcal{D}}$ [Deng and Zheng, 2021, Tzeng et al., 2017], with the intuition that the distribution gap impacts classification accuracy. In the following, we motivate our method by analyzing the failure of those methods based on feature-level distribution distance.

## 2.2  The failure of distribution distance

In the literature, distribution discrepancy has been considered as a key metric to predict the generalization performance of the model on unseen datasets [Deng and Zheng, 2021, Tzeng et al., 2017, Gao and Kleywegt, 2022, Sinha et al., 2017, Yu et al., 2022]. AutoEval [Deng and Zheng, 2021] estimates model performance by quantifying the domain gap in the feature space:

$$S(\mathcal{D}, \tilde{\mathcal{D}}) = d(\mathcal{D}, \tilde{\mathcal{D}}),$$

where $d(\cdot)$ denotes the distance function, such as Fréchet Distance [Dowson and Landau, 1982] or maximum mean discrepancy (MMD) [Gretton et al., 2006].

ProjNorm measures the distribution gap with the Euclidean distance in the parameter space:

$$S(\mathcal{D}, \tilde{\mathcal{D}}) = \|\theta - \tilde{\theta}\|_2,$$

where $\theta$ and $\tilde{\theta}$ denote the parameters fine-tuned on training data $\mathcal{D}$ and OOD data $\tilde{\mathcal{D}}$, respectively.

The underlying assumption is inherited from the conventional wisdom in domain adaptation, where a representation function that minimizes domain difference leads to higher accuracy in the target domain [Tzeng et al., 2014, Ganin and Lempitsky, 2015, Tzeng et al., 2017]. Yet, the relationship between distribution distance and test accuracy remains controversial. For example, the distribution shift may not change the model prediction if the classifier's outputs are insensitive to changes in the shifted features. There naturally arises a question: given a fixed model, does a larger distribution distance always lead to a lower test accuracy?

To verify the correlation between distribution distance and model performance, we compare the model performance on different OOD test sets of CIFAR-10C, using the ResNet-50 model trained on CIFAR-10. For each OOD test set, we calculate the distribution distances in the feature space $\mathcal{Z}$ via Fréchet distance [Dowson and Landau, 1982] and MMD [Gretton et al., 2006], respectively.

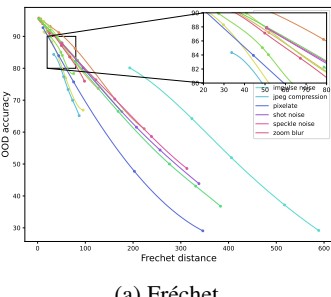

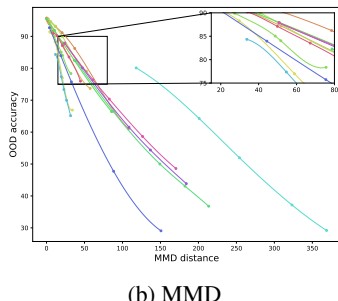

(a) Fréchet                                          (b) MMD

Figure 1: Distribution Gap Vs. OOD accuracy on CIFAR10-C via (a) Fréchet distance [Dowson and Landau, 1982] and (b) MMD [Gretton et al., 2006]. All colors indicate 14 types of corruption. The test accuracy varies significantly on shifted datasets with the same distribution distances.

Figure 1 presents the classification accuracy versus the distribution distance. The results show that, on different test datasets with similar distances to the train data in the feature space. the performance of the trained model varies significantly with both the two distance metrics. For example, calculated by Fréchet distance, the distribution gap varies only a small margin from 223.36 to 224.04, but the true accuracy experiences a large drop from 61.06% to 46.99%. Similar to MMD distance, when the distribution gap changes from 87.63 to 88.35, the OOD accuracy drops from 66.53% to 47.73%. The high variability in the model performance reveals that, **the distribution difference is not a reliable surrogate for generalization performance under distribution shifts**.

## 3   Proposed Method

In this section, we first introduce the intuition of our method with a natural example. Next, we characterize and provide quantitative measure on the desirable properties of feature representations for predicting OOD error. We then present the advantages of our proposed method.

### 3.1   Motivation

In representation learning, it is desirable to learn a separable feature space, where different classes are far away from each other and samples in each class form a compact cluster. Therefore, separability is viewed as an important characteristic to measure the feature quality. For example, one may train a nearest neighbor classifier over the learned representation using labeled data and regard its performance as the indicator. In this paper, we investigate how to utilize the characteristic of separability for estimating the model performance under distribution shifts, without access to ground-truth labels.

**Intuitive example**    In Figure 2, we present a t-SNE visualization of the representation for training and test data. In particular, we compare the separability of the learned representation on shifted datasets with various severity. While the feature representation of training data exhibits well-separated clusters, those of the shifted datasets are more difficult to differentiate and the cluster gaps are well correlated with the corruption severity, as well as the ground-truth accuracy. It implies that, a model with high classification accuracy is usually along with a well-separated feature distribution, and vice versa. With the intuition, we proceed by theoretically showing how the feature separability affects the classification performance.

**Theoretical explanation**    Recall that we obtain the model prediction by $y_i' = \arg\max f_\omega(z_i)$, where the intermediate feature $z_i = f_g(x_i)$. Let $P(y = i)$ denote the prior class probability of class $i$, and $p(x|y = i)$ denote the class likelihood, i.e., the conditional probability density of x given that it belongs to class $i$. Then the Bayes error [Young and Calvert, 1974, Devijver and Kittler, 1982, Webb and Copsey, 1990, Duda et al., 2006] can be expressed as:

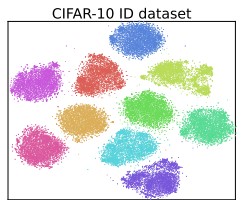 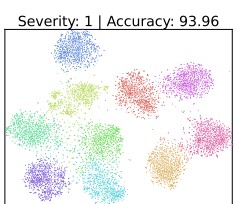 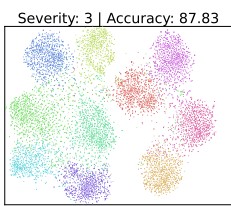 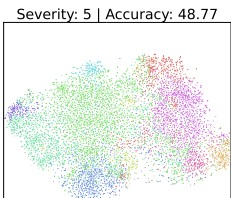

| CIFAR-10 ID dataset | Severity: 1 | Accuracy: 93.96 | Severity: 3 | Accuracy: 87.83 | Severity: 5 | Accuracy: 48.77 |

Figure 2: t-SNE visualization of feature representation on training and OOD test datasets (CIFAR-10C) with *contrast* corruption under different severity levels. The first left figure shows feature representation of the training dataset, while the rest of three figures illustrate those of the OOD datasets. From this figure, we observe that different clusters tends to be more separated as the corruption severity level gets smaller.

$$E_{\text{bayes}} = 1 - \sum_{i=1}^{k} \int_{C_i} P(y=i) \, p(\boldsymbol{x} \mid y=i) \, d\boldsymbol{x},$$

where $C_i$ is the region where class $i$ has the highest posterior. The Bayes error rate provides a lower bound on the error rate that can be achieved by any pattern classifier acting on derived features. However, the Bayes error is not so readily obtainable as class priors and class-conditional likelihoods are normally unknown. Therefore, many studies have focused on deriving meaningful upper bounds for the Bayes error [Devijver and Kittler, 1982, Webb and Copsey, 1990].

For a binary classification problem where $k = 2$, we have

$$E_{\text{bayes}} \leq \frac{2P(y=1)P(y=2)}{1 + P(y=1)P(y=2)\Delta}, \tag{2}$$

where $\Delta$ denotes the distance between features from the two classes [Devijver and Kittler, 1982, Tumer and Ghosh, 2003], such as *Mahalanobis distance* [Chandra et al., 1936] or *Bhattacharyya distance* [Fukunaga, 2013]. Based on the Bayes error for 2-class problems, the upper bound can be extended to the multi-class setting ($k > 2$) by the following equation [Garber and Djouadi, 1988].

$$E_{\text{bayes}}^{k} \leq \min_{\alpha \in \{0,1\}} \left( \frac{1}{k-2\alpha} \sum_{i=1}^{k} (1 - P(y=i)) E_{\text{bayes};i}^{k-1} + \frac{1-\alpha}{k-2\alpha} \right), \tag{3}$$

where $\alpha$ is an optimization parameter. With Equations 2 and 3, we obtain an upper bound of Bayes error, which is negatively correlated with the inter-class feature distances. Specifically, increasing the feature distance will result in a decrease in the upper bound of Bayes error. In this manner, we provide a mathematical intuition for the phenomenon shown in Figure 2. However, Computing the upper bound of Bayes error directly, as indicated above, is challenging due to its computational complexity. To circumvent this issue, we propose a straightforward metric as a substitute, which does not require access to label information.

## 3.2 Dispersion score

In our previous analysis, we show a strong correlation between the test accuracy and feature separability under different distribution shifts. To quantify the separability in the feature space $\mathcal{Z}$, we introduce *Dispersion score* that measures the inter-class margin without annotation information.

First, we allocate OOD instances $\{\tilde{\boldsymbol{x}}\}_{i=1}^{m}$ into different clusters $j$ based on the model predictions, i.e., their pseudo labels from the trained classifier $f_\omega$: $j = \tilde{y}_i' = \arg\max f_\omega(\boldsymbol{z}_i)$.

With these clusters, we compute the *Dispersion score* by the average distances between each cluster centroid $\tilde{\boldsymbol{\mu}}_j = \frac{1}{m_j} \sum_{i=1}^{m_j} \boldsymbol{z}_i \cdot \mathbb{1}\{\tilde{y}_i' = j\}$ and the center of all features $\bar{\boldsymbol{\mu}} = \frac{1}{m} \sum_{i=1}^{m} \boldsymbol{z}_i$, weighted by the sample size of each cluster $m_j$. Formally, the *Dispersion score* is defined as:

$$S(\tilde{\mathcal{D}}) = \frac{1}{k-1} \sum_{j=1}^{k} m_j \cdot \varphi(\bar{\boldsymbol{\mu}}, \tilde{\boldsymbol{\mu}}_j)$$

where $k-1$ is the degree of freedom and $\varphi$ denotes the distance function. With the weight $m_j$, the induced score receives a stronger influence from those larger clusters, i.e., the majority classes. This enables our method to be more robust to the long-tailed issue, which naturally arises in the unlabeled OOD data. In subsection 4.3, we explicitly show the advantage of our method in the class-imbalanced case and analyze the importance of the weight in Appendix 4.5.

In particular, we use square Euclidean distances to calculate the distance in the feature space $\mathcal{Z}$. So it converts to:

$$S(\tilde{\mathcal{D}}) = \log \frac{\sum_{j=1}^{k} m_j \cdot \|\bar{\boldsymbol{\mu}} - \tilde{\boldsymbol{\mu}}_j\|_2^2}{k-1} \tag{4}$$

Following the common practice [Jiang et al., 2019], we adopt a log transform on the final score, which corresponds to multiplicative combination of class statistics.

We summarize our approach in Appendix A. Notably, the Dispersion score derived from the feature separability offers several compelling advantages:

1. **Training data free**. The calculation procedure of our proposed score does not rely on the information of training data. Thus, our method is compatible with modern settings where models are trained on billions of images.

2. **Easy-to-use**. The computation of the Dispersion score only does forward propagation for each test instance once and does not require extra hyperparameter, training a new model, or updating the model parameters. Therefore, our method is easy to implement in real-world tasks and computational efficient, as demonstrated in Tables 2 and 3.

3. **Strong flexibility in OOD data**. Previous state-of-the-art methods, like ProjNorm [Yu et al., 2022], usually requires a large amount of OOD data for predicting the prediction performance. Besides, the class distribution of unlabeled OOD datasets might be severely imbalanced, which makes it challenging to estimate the desired test accuracy on balanced data. In Subsection 4.3, we will show the Dispersion score derived from the feature separability exhibits stronger flexibility and generality in sample size, class distribution, and partial label set (see Subsection 4.3 and Appendix E).

# 4 Experiments

In this subsection, we first compare the proposed score to existing training-free methods. Then, we provide an extensive comparison between our method and recent state-of-the-art method – ProjNorm. We then show the flexibility of our method under different settings of OOD test data. Additionally, we present an analysis of using ground-truth labels and K-means in Appendixes C and 4.6. Finally, we discuss the limitation of the proposed score for adversarial setting in Appendix F.

## 4.1 Experimental Setup

**Train datasets.** During training, we train models on the CIFAR-10, CIFAR-100 [Krizhevsky et al., 2009] and TinyImageNet [Le and Yang, 2015] datasets. Specifically, the train data of CIFAR-10 and CIFAR-100 contain 50,000 training images, which are allocated to 10 and 100 classes, respectively. The TinyImageNet dataset contains 100,000 $64 \times 64$ training images, with 200 classes.

**Out-of-distribution (OOD) datasets.** To evaluate the effectiveness of the proposed method on predicting OOD error at test time, we use CIFAR-10C and CIFAR-100C [Hendrycks and Dietterich, 2019], which span 19 types of corruption with 5 severity levels. For the testing of TinyImageNet, we use TinyImageNet-C [Hendrycks and Dietterich, 2019] that spans 15 types of corruption with 5 severity levels as OOD dataset. All the datasets with certain corruption and severity contain 10,000 images, which are evenly distributed in the classes.

**Evaluation metrics.** To measure the linear relationship between OOD error and designed scores, we use coefficients of determination ($R^2$) and Spearman correlation coefficients ($\rho$) as the evaluation metrics. For the comparison of computational efficiency, we calculate the average evaluation time ($T$) for each test set with certain corruption and severity.

**Training details.** During the training process, we train ResNet18, ResNet50 [He et al., 2016] and WRN-50-2 [Zagoruyko and Komodakis, 2016] on CIFAR-10, CIFAR-100 [Krizhevsky et al., 2009] and TinyImageNet [Le and Yang, 2015] with 20, 50 and 50 epochs, respectively. We use SGD with

Table 1: Performance comparison of training free approaches on CIFAR-10, CIFAR-100 and Tiny-ImageNet, where $R^2$ refers to coefficients of determination, and $\rho$ refers to Spearman correlation coefficients (higher is better). The best results are highlighted in **bold**.

| Dataset | Network | Rotation | | ConfScore | | Entropy | | AgreeScore | | ATC | | Fréchet | | Ours | |
|---|---|---|---|---|---|---|---|---|---|---|---|---|---|---|---|
| | | $R^2$ | $\rho$ | $R^2$ | $\rho$ | $R^2$ | $\rho$ | $R^2$ | $\rho$ | $R^2$ | $\rho$ | $R^2$ | $\rho$ | $R^2$ | $\rho$ |
| CIFAR 10 | ResNet18 | 0.822 | 0.951 | 0.869 | 0.985 | 0.899 | 0.987 | 0.663 | 0.929 | 0.884 | 0.985 | 0.950 | 0.971 | **0.968** | **0.990** |
| | ResNet50 | 0.835 | 0.961 | 0.935 | 0.993 | 0.945 | **0.994** | 0.835 | 0.985 | 0.946 | **0.994** | 0.858 | 0.964 | **0.987** | 0.990 |
| | WRN-50-2 | 0.862 | 0.976 | 0.943 | **0.994** | 0.942 | **0.994** | 0.856 | 0.986 | 0.947 | **0.994** | 0.814 | 0.973 | **0.962** | 0.988 |
| | Average | 0.840 | 0.963 | 0.916 | 0.991 | 0.930 | **0.992** | 0.785 | 0.967 | 0.926 | 0.991 | 0.874 | 0.970 | **0.972** | 0.990 |
| CIFAR 100 | ResNet18 | 0.860 | 0.936 | 0.916 | 0.985 | 0.891 | 0.979 | 0.902 | 0.973 | 0.938 | 0.986 | 0.888 | 0.968 | **0.952** | **0.988** |
| | ResNet50 | 0.908 | 0.962 | 0.919 | 0.984 | 0.884 | 0.977 | 0.922 | 0.982 | 0.921 | 0.984 | 0.837 | 0.972 | **0.951** | **0.985** |
| | WRN-50-2 | 0.924 | 0.970 | 0.971 | 0.984 | 0.968 | 0.981 | 0.955 | 0.977 | 0.978 | **0.993** | 0.865 | 0.987 | **0.980** | 0.991 |
| | Average | 0.898 | 0.956 | 0.936 | 0.987 | 0.915 | 0.983 | 0.927 | 0.982 | 0.946 | **0.988** | 0.864 | 0.976 | **0.962** | **0.988** |
| TinyImageNet | ResNet18 | 0.786 | 0.946 | 0.670 | 0.869 | 0.592 | 0.842 | 0.561 | 0.853 | 0.751 | 0.945 | 0.826 | 0.970 | **0.966** | **0.986** |
| | ResNet50 | 0.786 | 0.947 | 0.670 | 0.869 | 0.651 | 0.892 | 0.560 | 0.853 | 0.751 | 0.945 | 0.826 | 0.971 | **0.977** | **0.986** |
| | WRNt-50-2 | 0.878 | 0.967 | 0.757 | 0.951 | 0.704 | 0.935 | 0.654 | 0.904 | 0.635 | 0.897 | 0.884 | 0.984 | **0.968** | **0.986** |
| | Average | 0.805 | 0.959 | 0.727 | 0.920 | 0.650 | 0.890 | 0.599 | 0.878 | 0.693 | 0.921 | 0.847 | 0.976 | **0.970** | **0.987** |

Figure 3: OOD error prediction versus True OOD error on CIFAR-10 with ResNet50. We compare the performance of Dispersion Score with that of ProjNorm and Fréchet via scatter plots. Each point represents one dataset under certain corruption and certain severity, where different shapes represent different types of corruption, and darker color represents the higher severity level.

the learning rate of $10^{-3}$, cosine learning rate decay [Loshchilov and Hutter, 2016], a momentum of 0.9 and a batch size of 128 to train the model.

**The compared methods.** We consider 7 existing methods as benchmarks for predicting OOD error: *Rotation Prediction* (Rotation) [Deng et al., 2021], *Averaged Confidence* (ConfScore) [Hendrycks and Gimpel, 2016], *Entropy* [Guillory et al., 2021], *Agreement Score* (AgreeScore) [Jiang et al., 2021], *Averaged Threshold Confidence* (ATC) [Garg et al., 2022], *AutoEval* (Fréchet) [Deng and Zheng, 2021], and *ProjNorm* [Yu et al., 2022]. Rotation and AgreeScore predict OOD error from the view of unsupervised loss constructed by generating rotating instances and measuring output agreement of two independent models, respectively. ConfScore, Entropy, and ATC formulate scores by model predictions. ProjNorm measures the parameter-level difference between the models fine-tuned on train and OOD test data respectively, while Fréchet quantifies the distribution difference in the feature space between the training and test datasets. We present the related works in Appendix B.

## 4.2 Results

**Can Dispersion score outperform existing training-free approaches?** In Table 1, we present the performance of OOD error estimation on three model architectures and three datasets. We find that Dispersion Score dramatically outperforms existing training-free methods. For example, averaged across three architectures on TinyImageNet, our method leads to an increase of the $R^2$ from 0.847 to 0.970 – a 14.5% of relative improvement. In addition, Dispersion Score achieves consistently high performance over the three datasets with a $R^2$ higher than 0.950, while scores of other approaches such as Rotation varying from 0.787 to 0.924 are not stable. We observe a similar phenomenon on $\rho$, where the Entropy method achieves performance that is ranging from 0.842 to 0.994, while the performance of our method fluctuates around 0.988.

Table 2: Performance comparison between ProjNorm [Yu et al., 2022] and our Dispersion score on CIFAR-10, CIFAR-100 and TinyImageNet, where $R^2$ refers to coefficients of determination, $\rho$ refers to Spearman correlation coefficients (higher is better), and $T$ refers to average evaluation time (lower is better). The best results are highlighted in **bold**.

| Dataset | Network | ProjNorm | | | Ours | | |
|---|---|---|---|---|---|---|---|
| | | $R^2$ | $\rho$ | $T$ | $R^2$ | $\rho$ | $T$ |
| CIFAR 10 | ResNet18 | 0.936 | 0.982 | 179.616 | **0.968** | **0.990** | **10.980** |
| | ResNet50 | 0.944 | 0.989 | 266.099 | **0.987** | **0.990** | **11.259** |
| | WRN-50-2 | 0.961 | **0.989** | 575.888 | **0.962** | 0.988 | **11.017** |
| | Average | 0.947 | 0.987 | 326.201 | **0.972** | **0.990** | **11.085** |
| CIFAR 100 | ResNet18 | **0.979** | 0.980 | 180.453 | 0.952 | **0.988** | **6.997** |
| | ResNet50 | **0.988** | **0.991** | 262.831 | 0.953 | 0.985 | **11.138** |
| | WRN-50-2 | **0.990** | **0.991** | 605.616 | 0.980 | **0.991** | **12.353** |
| | Average | **0.985** | 0.987 | 349.63 | 0.962 | **0.988** | **10.163** |
| TinyImageNet | ResNet18 | **0.970** | 0.981 | 182.127 | 0.966 | **0.986** | **7.039** |
| | ResNet50 | **0.979** | 0.987 | 264.651 | 0.977 | **0.990** | **13.938** |
| | WRN-50-2 | 0.965 | 0.983 | 590.597 | **0.968** | **0.986** | **11.235** |
| | Average | **0.972** | 0.984 | 345.792 | 0.970 | **0.987** | **10.737** |

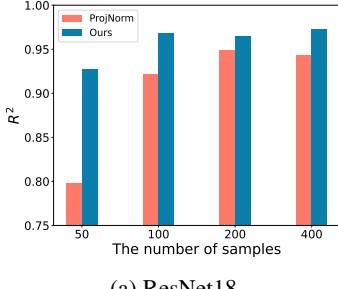

(a) ResNet18

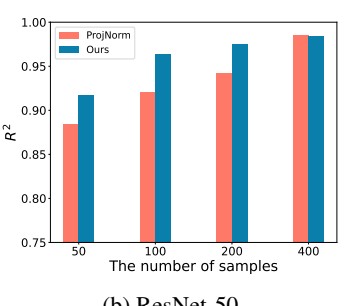

(b) ResNet-50

Figure 4: Prediction performance $R^2$ vs. sample size of OOD data (subsets of CIFAR-10C) with (a) ResNet18 and (b) ResNet50.

**Dispersion score is superior to ProjNorm.** In Table 2, we compare our method to the recent state-of-the-art method – ProjNrom [Yu et al., 2022] in both prediction performance and computational efficiency. The results illustrate that Dispersion Score could improve the prediction performance over ProjNorm with a meaningful margin. For example, with trained models on CIFAR-10 dataset, using Dispersion score achieves a $R^2$ of 0.953, much higher than the average performance of ProjNorm as 0.873. On CIFAR-100, our method also achieves comparable (slightly better) performance with ProjNorm (0.961 vs. 0.948). Besides, Dispersion score obtains huge advantages to ProjNorm in computational efficiency. Using WRN-50-2, ProjNorm takes an average of 575 seconds for estimating the performance of each OOD test dataset, while our method only requires 11 seconds. Since the computation of Dispersion score does not needs to update model parameters or utilize the training data, our method enables to predict OOD error with large-scale models trained on billions of images.

To further analyze the advantage of our method, we present in Figure 3 the scatter plots for Fréchet, ProjNorm and Dispersion Score on CIFAR-10C with ResNet50. From the figure, we find that Dispersion Score estimates OOD errors linearly w.r.t. true OOD errors in all cases. In contrast, we observe that those methods based on distribution difference tends to fail when the classification error is high. This phenomenon clearly demonstrates the reliable and superior performance of the Dispersion score in predicting generalization performance under distribution shifts.

## 4.3 Flexibility in OOD data

In previous analysis, we show that Dispersion score can outperform existing methods on standard benchmarks, where the OOD test datasets contain sufficient instances and have a balanced class distribution. In reality, the unlabeled OOD data are naturally imperfect, which may limit the performance of predicting OOD error. In this part, we verify the effectiveness of our method with long-tailed data and small data, compared to ProjNorm [Yu et al., 2022].

Table 3: Summary of prediction performance on **Imbalanced** CIFAR-10C and CIFAR-100C [Hendrycks and Dietterich, 2019], where $R^2$ refers to coefficients of determination, $\rho$ refers to Spearman correlation coefficients (higher is better), and $T$ refers to average evaluation time (lower is better). The best results are highlighted in **bold**. More results can be found in Appendix D.

| Dataset | Network | ProjNorm | | | Ours | | |
|---------|---------|----------|----------|----------|----------|----------|----------|
| | | $R^2$ | $\rho$ | $T$ | $R^2$ | $\rho$ | $T$ |
| CIFAR 10 | ResNet18 | 0.799 | 0.968 | 204.900 | **0.959** | **0.982** | **2.076** |
| | ResNet50 | 0.897 | 0.980 | 430.406 | **0.968** | **0.982** | **3.295** |
| | WRN-50-2 | 0.922 | 0.978 | 561.611 | **0.932** | **0.978** | **2.970** |
| | Average | 0.873 | 0.973 | 398.972 | **0.953** | **0.980** | **2.780** |
| CIFAR 100 | ResNet18 | 0.886 | 0.968 | 210.05 | **0.941** | **0.982** | **1.864** |
| | ResNet50 | **0.980** | **0.988** | 433.860 | 0.956 | 0.982 | **2.974** |
| | WRN-50-2 | 0.978 | 0.982 | 768.883 | **0.986** | **0.994** | **3.242** |
| | Average | 0.948 | 0.980 | 470.931 | **0.961** | **0.986** | **2.693** |

**Class imbalance.** We first evaluate the prediction performance under class imbalanced setting. In particular, given a trained model, we aim to estimate its balanced accuracy under distribution shift while we only have access to a long-tailed test data, where a few classes (majority classes) occupy most of the data and most classes (minority classes) are under-represented. We conduct experiments on CIFAR-10C and CIFAR-100C with imbalance rate 100.

Our results in Table 3 show that Dispersion Score achieves better performance than ProjNorm [Yu et al., 2022] under class imbalance. For example, our approach achieves an average $R^2$ of 0.953 on CIFAR-10C with 2.780 seconds, while ProjNorm obtains an $R^2$ of 0.873 and takes 398.972 seconds. In addition, the performance of our method is more stable across different model architectures and datasets with the $R^2$ ranging from 0.932 to 0.986 than ProjNorm (0.799 - 0.980). Overall, Dispersion score maintains reliable and superior performance even when the OOD test set are class imbalanced.

**Sampling efficiency.** During deployment, it can be challenging to collect instances under a specific distribution shift. However, existing state-of-the-art methods [Yu et al., 2022] normally require a sufficiently large test dataset to achieve meaningful results in predicting OOD error. Here, we further validate the sampling efficiency of Dispersion score, compared with ProjNorm.

Figure 4 presents the performance comparison between Dispersion Score and ProjNorm on subsets of CIFAR10C with various sample sizes. The results show that our method can achieve excellent performance even when only 50 examples are available in the OOD data. In contrast, the performance of ProjNorm decreases sharply with the decrease in sample size. The phenomenon shows that Dispersion score is more efficient in exploiting the information of OOD instances.

### 4.4 Intra-class compactness Vs Inter-class dispersion

In Section 3, we empirically show that feature separability is naturally tied with the final accuracy and propose an effective score based on inter-class dispersion. However, the connection between intra-class compactness and generalization performance is still a mystery. In this analysis, we show that compactness is not a good indicator of OOD accuracy. Specifically, we define the compactness score:

$$S(\tilde{\mathcal{D}}) = -\log \frac{\sum_{j=1}^{k} \sum_{i=1}^{m_j} ||\boldsymbol{z}_i - \tilde{\boldsymbol{\mu}}_j||^2}{n - k}$$

where $\tilde{\boldsymbol{\mu}}_j$ denotes the centroid of the cluster $j$ that the instance $\boldsymbol{z}_i$ belongs to. Therefore, the compactness score can measure the clusterability of the learned features, i.e., the average distance between each instance and its cluster centroid. Intuitively, a high compactness score may correspond to a

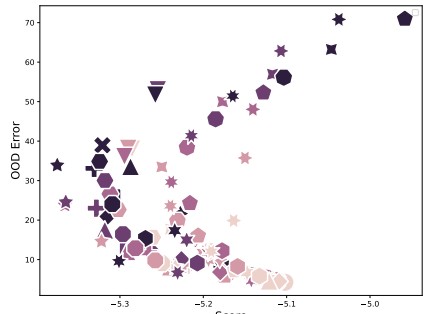

Figure 5: Compactness vs. test error on CIFAR-10C with ResNet50.

well-separated feature distribution, which leads to high test accuracy. Surprisingly, in Figure 5, we find that the compactness score is largely irrelevant to the final OOD error, showing that it is not an effective indicator for predicting generalization performance under distribution shifts.

## 4.5 The importance of the weight

As introduced in Section 3, Dispersion Score can be viewed as a weighted arithmetic mean of the distance from centers of each class to the center of the whole samples in the feature space, where the weight is the total number of samples in the corresponding class. To verify the importance of the weight in long-tailed setting, we consider a variant that removes the weight:

$$S(\tilde{\mathcal{D}}) = \frac{1}{k-1} \sum_{j=0}^{k} \varphi(\bar{\boldsymbol{\mu}}, \tilde{\boldsymbol{\mu}}_j).$$

The results are shown in Table 4, where we compare performances of Dispersion score and the variant without weight respectively under **imbalanced** CIFAR-10C and CIFAR-100C with ResNet10 and ResNet50. From this table, we could observe that the weight enhance the robustness significantly in long-tail conditions.

Table 4: Summary of prediction performance on **Imbalanced** CIFAR-10C and CIFAR-100C. The best results are highlighted in **bold**.

| Dataset | Network | w/o Weights | | w/ Weights | |
|---|---|---|---|---|---|
| | | $R^2$ | $\rho$ | $R^2$ | $\rho$ |
| CIFAR 10 | ResNet18 | 0.675 | 0.930 | **0.959** | **0.982** |
| | ResNet50 | 0.748 | 0.948 | **0.968** | **0.982** |
| | Average | 0.712 | 0.939 | **0.978** | **0.990** |
| CIFAR 100 | ResNet18 | 0.595 | 0.838 | **0.941** | **0.982** |
| | ResNet50 | 0.395 | 0.733 | **0.956** | **0.982** |
| | Average | 0.494 | 0.785 | **0.952** | **0.986** |

## 4.6 K-means vs. Pseudo labels.

While our Dispersion score derived from pseudo labels has demonstrated strong promise, a question arises: *can a similar effect be achieved by alternative clustering methods?* In this ablation, we show that labels obtained by K-means [Lloyd, 1982, MacQueen, 1967] does not achieve comparable performance with pseudo labels obtained from the trained classifier. In particular, we allocate instances into clusters by using K-means instead of pseudo labels from the classifier.

We present the performance comparison of our method and the variant of K-means in Figure 6. The results show that our Dispersion score performs better than the K-means variant and the gap is enlarged with more classes. On the other hand, the variant of K-means does not require a

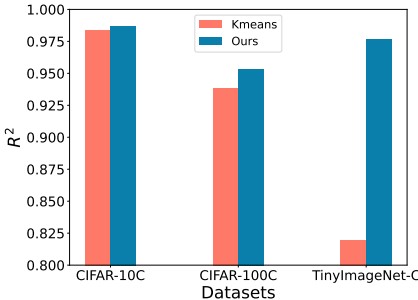

Figure 6: Compare with K-means.

classifier, i.e., the linear layer in the trained model, which enables to evaluate the OOD performance of representation learning methods, e.g., self-supervised learning.

## 5 Conclusion

In this paper, we introduce Dispersion score, a simple yet effective indicator for predicting the generalization performance on OOD data without labels. We show that Dispersion score is strongly correlated to the OOD error and achieves consistently better performance than previous methods under various distribution shifts. Even when the OOD datasets are class imbalanced or have limit number of instances, our method maintains a high prediction performance, which demonstrates the strong flexibility of dispersion score. This method can be easily adopted in practical settings. It is straightforward to implement with trained models with various architectures, and does not require access to the training data. Thus, our method is compatible with large-scale models that are trained on billions of images. We hope that our insights inspire future research to further explore the feature separability for predicting OOD error.

# 6   Acknowledgements

This research is supported by the Ministry of Education, Singapore, under its Academic Research Fund Tier 1 (RG13/22). Hongxin Wei gratefully acknowledges the support of Center for Computational Science and Engineering at Southern University of Science and Technology for our research. Lei Feng is supported by the National Natural Science Foundation of China (Grant No. 62106028), Chongqing Overseas Chinese Entrepreneurship and Innovation Support Program, Chongqing Artificial Intelligence Innovation Center, CAAI-Huawei MindSpore Open Fund, and Openl Community (https://openi.pcl.ac.cn).

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

# A    The calculation of Dispersion Score

Our proposed approach, Dispersion Score, can be calculated as shown in Algorithm 1.

---

**Algorithm 1** OOD Error Estimation via Dispersion Score

---

**Input:** OOD test dataset $\tilde{\mathcal{D}} = \{\tilde{x}_i\}_{i=1}^m$, a trained model $f$ (feature extractor $f_g$ and classifier $f_\omega$)
**Output:** The dispersion score
**for** each OOD instance $\tilde{x}_i$ **do**
    Obtain feature representation via $\tilde{z}_i = f_g(\tilde{x}_i)$.
    Obtain pseudo labels via $\tilde{y}_i' = \arg\max f_\omega(z_i)$
**end for**
Calculate cluster centroids $\{\tilde{\mu}_j\}_{j=1}^k$ using pseudo labels $\tilde{y}_i'$, with $\tilde{\mu}_j = \frac{1}{m_j}\sum_{i=1}^{m_j} z_i \cdot \mathbb{1}\{\tilde{y}_i' = j\}$
Calculate the feature center of all instances by $\bar{\mu} = \frac{1}{m}\sum_{i=1}^m z_i$
Calculate Dispersion Score $S(\tilde{\mathcal{D}})$ via Equation (4)

---

# B    Related work

**Predicting generalization.** Since the generalization capability of deep networks under distribution shifts is a mysterious desideratum, a surge of researches pay attention to estimate the generalization capability from two directions.

1) Some works aim to measure generalization gap between training and test accuracy with only training data [Corneanu et al., 2020, Jiang et al., 2019, Neyshabur et al., 2017, Unterthiner et al., 2020, Yak et al., 2019, Martin and Mahoney, 2020]. For example, the model-architecture-based method [Corneanu et al., 2020] summarizes the persistent topological map of a trained model to formulate its inner-working function, which represents the generalization gap. Margin distribution [Jiang et al., 2019] measures the gap by gauging the distance between training examples and the decision boundary. However, those methods are designed for the identical distribution between the training and test dataset, being vulnerable to distribution shift.

2) Some studies try to estimate generalization performance on a specific OOD test dataset without annotation during evaluation. Many of them utilize softmax outputs of the shifted test dataset to form a quantitative indicator of OOD error [Guillory et al., 2021, Jiang et al., 2021, Guillory et al., 2021, Garg et al., 2022]. However, those methods are unreliable across diverse distribution shifts due to the overconfidence problem [Wei et al., 2022]. Another popular direction considers the negative correlation between distribution difference and model's performance in the space of features [Deng and Zheng, 2021] or parameters [Yu et al., 2022]. Nevertheless, common distribution distances practically fail to induce stable error estimation under distribution shift [Guillory et al., 2021], and those methods are usually computationally expensive. Unsupervised loss such as agreement among multiple classifiers [Jiang et al., 2021, Madani et al., 2004, Platanios et al., 2016, 2017] and data augmentation [Deng et al., 2021] is also employed for OOD error prediction, which requires specific model structures during training. In this work, we focus on exploring the connection between feature separability and generalization performance under distribution shift, which is training-free and does not have extra requirements for datasets and model architectures.

**Exploring Feature distribution in deep learning.** In the literature, feature distribution has been widely studied in domain adaptation [Ben-David et al., 2006, Pan et al., 2010, Zhuang et al., 2015, Tzeng et al., 2017], representation learning [Bengio et al., 2013, HaoChen et al., 2021, Ming et al., 2023, Huang et al., 2021], OOD generalization [Li et al., 2018, Chen et al., 2021, Wang et al., 2021], and noisy-label learning [Zhu et al., 2021, 2022]. Domain adaptation methods usually learn a domain-invariant feature representation by narrowing the distribution distance between the two domains with certain criteria, such as maximum mean discrepancy (MMD) [Pan et al., 2010], Kullback-Leibler (KL) divergence [Zhuang et al., 2015], central moment discrepancy (CMD) [Zellinger et al., 2017], and Wasserstein distance [Lee and Raginsky, 2017]. InfoNCE [Huang et al., 2021] shows a key factor of contrastive learning that the distance between class centers should be large enough. In learning with noisy labels, it has been shown that the feature clusterability can be used to estimate the transition matrix [Zhu et al., 2021]. To the best of our knowledge, we are the first to analyze the connection between feature separability and the final accuracy on OOD data.

## C Sensitivity analysis: pseudo labels

Here, we conduct a sensitivity analysis by using ground-truth labels in our method. Table 5 illustrates that the performance with pseudo labels is comparable with the performance using ground-truth labels. This phenomenon is consistent with the previous method - ProjNorm [Yu et al., 2022], shown in Table 8 of their paper. The reason behind this could be that the trained model is capable of identifying certain semantic information from most corrupted examples, thereby retaining their representations in a cluster. Thus, the separability of feature clusters can serve as an indicator of the final prediction performance for corruption perturbations. Additionally, we provide a failure case of feature clusters separability for adversarial perturbations in Appendix F.

Table 5: Comparison of Dispersion Score with pseudo labels and true labels on CIFAR10, CIFAR100 and TinyImageNet. The best results are highlighted in **bold**.

| Dataset | Network | Pseudo labels | | True labels | |
|---|---|---|---|---|---|
| | | $R^2$ | $\rho$ | $R^2$ | $\rho$ |
| CIFAR 10 | ResNet18 | 0.968 | **0.990** | **0.979** | 0.989 |
| | ResNet50 | **0.987** | 0.990 | 0.985 | **0.991** |
| | WRN-50-2 | **0.961** | **0.988** | 0.945 | 0.987 |
| | Average | **0.972** | **0.990** | 0.970 | 0.989 |
| CIFAR 100 | ResNet18 | **0.952** | 0.988 | 0.915 | **0.989** |
| | ResNet50 | 0.953 | 0.985 | **0.959** | **0.989** |
| | WRN-50-2 | **0.980** | 0.991 | 0.978 | **0.995** |
| | Average | **0.962** | 0.988 | 0.950 | **0.991** |
| TinyImageNet | ResNet18 | **0.966** | **0.986** | 0.937 | 0.985 |
| | ResNet50 | **0.977** | 0.990 | 0.954 | **0.995** |
| | WRN-50-2 | 0.968 | 0.986 | **0.977** | **0.994** |
| | Average | **0.970** | 0.987 | 0.956 | **0.991** |

## D More results on class-imbalance settings

This section provides elaborated outcomes of training-free benchmarks under the setting of class imbalance, serving as a complement to the results presented in Table 3.

Table 6: Summary of prediction performance on **Imbalanced** CIFAR-10C and CIFAR-100C for training-free benchmarks, where $R^2$ refers to coefficients of determination, and $\rho$ refers to Spearman correlation coefficients (higher is better)

| Dataset | Network | Rotation | | ConfScore | | Entropy | | AgreeScore | | ATC | | Fréchet | |
|---|---|---|---|---|---|---|---|---|---|---|---|---|---|
| | | $R^2$ | $\rho$ | $R^2$ | $\rho$ | $R^2$ | $\rho$ | $R^2$ | $\rho$ | $R^2$ | $\rho$ | $R^2$ | $\rho$ |
| CIFAR 10 | ResNet18 | 0.767 | 0.922 | 0.823 | 0.965 | 0.841 | 0.969 | 0.669 | 0.922 | 0.830 | 0.966 | 0.966 | 0.983 |
| | ResNet50 | 0.787 | 0.946 | 0.870 | 0.975 | 0.887 | 0.977 | 0.765 | 0.953 | 0.883 | 0.975 | 0.916 | 0.975 |
| | WRN-50-2 | 0.829 | 0.968 | 0.915 | 0.986 | 0.913 | 0.986 | 0.823 | 0.972 | 0.922 | 0.986 | 0.866 | 0.977 |
| | Average | 0.794 | 0.945 | 0.869 | 0.976 | 0.880 | 0.977 | 0.752 | 0.949 | 0.878 | 0.976 | 0.916 | 0.979 |
| CIFAR 100 | ResNet18 | 0.769 | 0.944 | 0.872 | 0.988 | 0.840 | 0.985 | 0.858 | 0.979 | 0.905 | 0.988 | 0.905 | 0.972 |
| | ResNet50 | 0.847 | 0.964 | 0.875 | 0.986 | 0.826 | 0.978 | 0.832 | 0.973 | 0.880 | 0.986 | 0.855 | 0.979 |
| | WRN-50-2 | 0.930 | 0.981 | 0.976 | 0.993 | 0.980 | 0.993 | 0.944 | 0.981 | 0.981 | 0.994 | 0.889 | 0.988 |
| | Average | 0.849 | 0.963 | 0.908 | 0.989 | 0.882 | 0.985 | 0.878 | 0.978 | 0.922 | 0.989 | 0.883 | 0.980 |

## E Partial OOD error prediction

In previous experiments, a common assumption is that the test set contains instances from all classes. To further explore the flexibility of our method on OOD test set, we introduce a new setting called *partial OOD error prediction*, where the label space for the test set is a subset of the label space for the training data.

Here, we train ResNet18 and ResNet50 on both CIFAR-10 and CIFAR-100 with 10 and 100 categories, respectively. Different from previous settings, we evaluate the prediction performance on CIFAR-10C and CIFAR-100C with the first 50% of categories. The numerical results are shown in Tabel 7.

Table 7: Summary of prediction performance on **partial sets** of CIFAR-10C and CIFAR-100C, where $R^2$ refers to coefficients of determination, and $\rho$ refers to Spearman correlation coefficients (higher is better). The best results are highlighted in **bold**.

| Dataset | Network | Rotation | | ConfScore | | Entropy | | AgreeScore | | ATC | | Frechet | | ProjNorm | | Ours | |
|---|---|---|---|---|---|---|---|---|---|---|---|---|---|---|---|---|---|
| | | $R^2$ | $\rho$ | $R^2$ | $\rho$ | $R^2$ | $\rho$ | $R^2$ | $\rho$ | $R^2$ | $\rho$ | $R^2$ | $\rho$ | $R^2$ | $\rho$ | $R^2$ | $\rho$ |
| CIFAR 10 | ResNet18 | 0.578 | 0.896 | 0.795 | 0.982 | 0.826 | 0.984 | 0.615 | 0.931 | 0.802 | 0.981 | 0.842 | 0.941 | 0.770 | 0.968 | **0.935** | **0.985** |
| | ResNet50 | 0.719 | 0.939 | 0.885 | **0.993** | 0.892 | **0.993** | 0.787 | 0.976 | 0.887 | **0.993** | 0.757 | 0.953 | 0.856 | 0.967 | **0.950** | 0.992 |
| | Average | 0.649 | 0.918 | 0.841 | 0.987 | 0.859 | **0.989** | 0.701 | 0.954 | 0.845 | 0.987 | 0.800 | 0.947 | 0.813 | 0.968 | **0.942** | 0.988 |
| CIFAR 100 | ResNet18 | 0.876 | 0.946 | 0.922 | 0.985 | 0.902 | 0.980 | 0.904 | 0.971 | **0.943** | **0.986** | 0.894 | 0.972 | 0.770 | 0.968 | 0.935 | 0.985 |
| | ResNet50 | 0.923 | 0.967 | 0.917 | 0.980 | 0.890 | 0.975 | 0.915 | 0.976 | 0.932 | 0.983 | 0.837 | 0.978 | 0.856 | 0.967 | **0.950** | **0.992** |
| | Average | 0.899 | 0.956 | 0.920 | 0.983 | 0.896 | 0.978 | 0.909 | 0.973 | 0.938 | 0.984 | 0.866 | 0.975 | 0.813 | 0.968 | **0.942** | **0.989** |

From the results, we could observe that our method is more robust than existing methods in the setting of partial OOD error prediction. For example, ProjNorm suffers from the incomplete test dataset during the self-training process, with a dramatic drop from around 0.950 to around 0.810 for $R^2$ of CIFAR-10C and CIFAR-100C on average. Contrastively, our method still achieves high accuracy in predicting OOD errors, maintaining an average $R^2$ value of 0.950.

# F    Adversarial vs. Corruption robustness.

In previous analysis, we show the superior performance of dispersion score on predicting the accuracy on OOD test sets with different corruptions. Here, we surprisingly find that feature dispersion can effectively demonstrate the difference between adversarial and corruption robustness.

Table 8: Prediction performance measured by MSE against adversarial attack of different methods. The linear regression model is estimated on CIFAR-10C, and is used to predict the adversarial examples with perturbation size $\epsilon$ varying from 0.25 to 8.0. "True Dispersion" refers to the dispersion score with feature normalization using ground-truth labels. The best results are highlighted in **bold**.

| | ConfScore | Entropy | ATC | ProjNorm | Dispersion | True Dispersion |
|---|---|---|---|---|---|---|
| CIFAR-10 | 0.933 | 0.892 | 0.906 | 0.847 | 1.359 | **0.483** |

Table 8 shows the prediction performance of different methods under adversarial attacks. "True Dispersion" refers to the dispersion score with feature normalization using ground-truth labels. In particular, we generate adversarial samples attacked by projected gradient descent (PGD) using untargeted attack [Kurakin et al., 2016] on the test set of CIFAR-10 with 10 steps and perturbation size $\epsilon$ ranging from 0.25 to 8.0. While the vanilla Dispersion score leads to poor performance, we note that the variant of Dispersion score with ground-truth labels performs much better than previous methods. This phenomenon is different from the conclusion of the sensitivity analysis of pseudo labels in predicting corruption robustness (See Appendix C), where the variant of true labels cannot outperform our method.

To understand the reasons behind the performance disparity of feature dispersion between adversarial and corruption robustness, we present the t-SNE visualization of features for adversarial attack of CIFAR-10 test set with various perturbation sizes in Figure 7. Compared to Figure 3, the results indicate that adversarial perturbations increase the distance between different clusters, whereas corruption perturbations decrease the separability of the clusters. In other words, adversarial perturbations decrease the test accuracy in a different way: assigning instances to the wrong groups and enlarging the distance among those groups. Therefore, feature dispersion using pseudo labels cannot be an effective method in the adversarial setting. We hope this insight can inspire specific designed methods based on feature dispersion for predicting adversarial errors in the future.

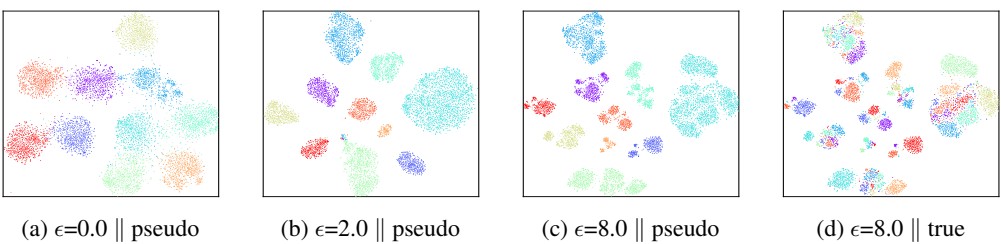

| (a) $\epsilon$=0.0 ‖ pseudo | (b) $\epsilon$=2.0 ‖ pseudo | (c) $\epsilon$=8.0 ‖ pseudo | (d) $\epsilon$=8.0 ‖ true |

Figure 7: t-SNE visualization of feature representation on adversarial attack of CIFAR-10 test set with perturbation size $\epsilon$ ranging from 0.25 to 8.0.

## G    Performance on realistic datasets

To verify the effectiveness of Dispersion Score on realistic datasets, we conduct experiments on PACS [Li et al., 2017], Office-31 [Saenko et al., 2010] and Office-Home [Venkateswara et al., 2017] with ResNet-18, ResNet-50 and WRN-50-2 using normalization. Table 9 is the numerical results, from which we can observe that our method outperforms the other baselines on datasets with natural shifts.

Table 9: Performance comparison of all approaches on PACS, Office-31 and Office-Home, where $R^2$ refers to coefficients of determination, and $\rho$ refers to Spearman correlation coefficients (higher is better). The best results are highlighted in **bold**.

| Dataset | Network | Rotation | | ConfScore | | Entropy | | AgreeScore | | ATC | | Fréchet | | ProjNorm | | Dispersion | |
|---|---|---|---|---|---|---|---|---|---|---|---|---|---|---|---|---|---|
| | | $R^2$ | $\rho$ | $R^2$ | $\rho$ | $R^2$ | $\rho$ | $R^2$ | $\rho$ | $R^2$ | $\rho$ | $R^2$ | $\rho$ | $R^2$ | $\rho$ | $R^2$ | $\rho$ |
| PACS | ResNet18 | 0.823 | **0.895** | 0.595 | 0.755 | 0.624 | 0.755 | 0.624 | 0.832 | 0.514 | 0.650 | 0.624 | 0.804 | 0.161 | 0.420 | **0.843** | 0.846 |
| | ResNet50 | **0.861** | **0.923** | 0.071 | 0.070 | 0.062 | 0.056 | 0.463 | 0.622 | 0.192 | 0.266 | 0.463 | 0.622 | 0.245 | 0.587 | 0.827 | 0.867 |
| | WRN-50-2 | 0.865 | 0.902 | 0.646 | 0.678 | 0.629 | 0.671 | 0.377 | 0.858 | 0.753 | 0.832 | 0.558 | 0.832 | 0.475 | 0.650 | **0.896** | **0.937** |
| | Average | 0.850 | 0.907 | 0.437 | 0.501 | 0.438 | **0.494** | 0.488 | 0.771 | 0.486 | 0.583 | 0.549 | 0.753 | 0.294 | 0.552 | **0.855** | **0.883** |
| Office-31 | ResNet18 | 0.753 | 0.943 | 0.470 | 0.829 | 0.322 | 0.714 | 0.003 | 0.086 | **0.844** | **0.943** | 0.144 | 0.257 | 0.099 | 0.429 | 0.834 | **0.943** |
| | ResNet50 | 0.371 | 0.829 | 0.486 | 0.829 | 0.355 | 0.829 | 0.012 | 0.464 | 0.533 | 0.486 | 0.035 | 0.257 | 0.241 | 0.429 | **0.878** | **0.943** |
| | WRN-50-2 | 0.578 | 0.600 | 0.525 | 0.714 | 0.425 | 0.714 | 0.003 | 0.257 | 0.405 | 0.943 | 0.035 | 0.143 | 0.147 | 0.143 | **0.798** | **0.829** |
| | Average | 0.567 | 0.790 | 0.936 | 0.494 | 0.790 | 0.367 | 0.752 | 0.006 | 0.269 | **0.594** | 0.790 | 0.219 | 0.162 | 0.333 | **0.836** | **0.905** |
| Office-Home | ResNet18 | **0.823** | **0.930** | 0.795 | 0.909 | 0.762 | 0.881 | 0.055 | 0.147 | 0.571 | 0.615 | 0.606 | 0.755 | 0.065 | 0.203 | 0.821 | 0.811 |
| | ResNet50 | **0.851** | **0.944** | 0.770 | 0.895 | 0.742 | 0.853 | 0.027 | 0.217 | 0.487 | 0.734 | 0.607 | 0.685 | 0.169 | 0.476 | 0.841 | 0.860 |
| | WRNt-50-2 | 0.823 | 0.958 | 0.742 | 0.874 | 0.696 | 0.846 | 0.132 | 0.406 | 0.384 | 0.643 | 0.589 | 0.706 | 0.173 | 0.531 | **0.897** | **0.937** |
| | Average | 0.832 | 0.944 | 0.769 | 0.893 | 0.734 | 0.860 | 0.071 | 0.256 | 0.481 | 0.664 | 0.601 | 0.716 | 0.135 | 0.403 | **0.853** | **0.869** |

