# OpenReview forum: "On the Importance of Feature Separability in Predicting Out-Of-Distribution Error"
_NeurIPS.cc/2023/Conference — NeurIPS 2023 poster_

### Official Review · Reviewer_ALm9 · 2023-06-17

**Soundness:** 2 fair
**Presentation:** 2 fair
**Contribution:** 2 fair
**Rating:** 6
**Confidence:** 4

**Summary:**

This paper proposes an easy-to-use dataset-level method for predicting OOD scores. The authors analyze two desiderata in representation learning: high inter-class dispersion and high intra-class compactness. Through some experiments on CIFAR and TinyImageNet, the authors reveal that the inter-class dispersion is strongly correlated with the OOD performance while intra-class compactness does not really correlate with the OOD accuracy.

**Strengths:**

1. The proposed method is well-motivated and explained. The authors first claim that MMD and Fr\'echet distance is not good surrogates for  OOD error prediction and suggest using dispersion score instead. Figure 3 indicates that the dispersion score is indeed better than conventionally used distance.

2. The proposed method is easy-of-use and training-free. It is also flexible to different OOD data in sample size and class distributions.

3. The method outperforms previous approaches in most benchmarks.

**Weaknesses:**

1. The experiments are all conducted in elementary and simple experiments settings, i.e., the OOD dataset is set to be some augmentation and corruptions applied on the ID set. This is not really the real-world OOD benchmark setting as the corrupted OOD dataset still has some class-overlapping information with the ID set. Can it apply to the standard OOD benchmark in CIFAR and ImageNet? For example, the authors could use ImageNet-1k as ID and iNaturalist, Places, Textures, and SUN as OOD. With the current experimental setting with simple data augmentation, it is really hard to judge whether the method can be useful for real-world usage.

2. It would be much more interesting to have non-overlapping ID and OOD dataset in experiments. For example, the authors can train model on CIFAR100 and use CIFAR10C as OOD or vice verse (CIFAR10 as ID and CIFAR100 as OOD). It also meets more real-world setting as CIFAR10/CIFAR100 is a commonly used OOD benchmark.

3. Though the intra-class compactness alone is not useful. Would it be better if the intra-class compactness is combined with the dispersion? Would it be an interesting ablation study?


**Questions:**

Please see weaknesses and limitations.

**Limitations:**

My main concern is still the simple experiment setting. In my opinion, it would make the proposed method useful only if the OOD benchmark is replaced with real-world ones. I suggest the authors verify the proposed approach in standard CIFAR and ImageNet-1k benchmarks as done in [1,2,3].  Of course, it is not necessary to plot the curve of accuracy versus distance as done in Figure 1 and Figure 3 (a single correlation value would be sufficient), but it would be important to show the method of predicting OOD error is useful in practice especially given that the method does not rely on the class distribution.

[1] React: Out-of-distribution detection with rectified activations. NeurIPS21

[2] On the importance of gradients for detecting distributional shifts in the wild, NeurIPS21

[3] RankFeat: Rank-1 Feature Removal for Out-of-distribution Detection. NeurIPS22

---

> ### Author Rebuttal · Authors · 2023-08-09
>
> 1. **About the OOD setting in the experiments**
>
> Thank you for pointing out the potential for misunderstanding. We would like to clarify that there are different definitions of "OOD" in various related areas. In OOD detection, OOD examples are those test samples with different label space from the training set (semantic shift). In OOD generalization [1] and OOD error estimation, we use "OOD" to describe examples with covariant shifts or concept shifts, where their ground-truth labels are included in the label set. The difference between these two settings lies in **whether the label space is shared between the training and test data**. Therefore, it is common practice to use different "OOD" test sets in OOD detection and OOD error estimation. To avoid potential misunderstandings, we will clearly describe the OOD setting in the problem statement (Subsection 2.1) of the final version.
>
> Moreover, it is meaningless to calculate the accuracy in an OOD test set with semantic shifts, where the ground-truth labels of all instances are out of the label space. To demonstrate the robustness of Dispersion score in complex settings, we conduct experiments on a more realistic setting, where parts of test samples are with semantic shifts and others are with covariant/concept shifts. The goal is to estimate the performance on those examples with covariant/concept shifts. We show the results in the Table 2 (see the attached pdf).
>
> Specifically, we inject 10% extra examples from unseen classes (drawn from [300K Random Images](https://github.com/hendrycks/outlier-exposure) and CIFAR-100) on ResNet18. This table shows that our method outperforms all training-free benchmarks, and is comparable with training method, ProjNorm (using ).
>
> In addition, to demonstrate effectiveness of Dispersion Score under natural distribution shifts, we also conduct experiments on some domain-adaptation/ domain-generalizaition datasets, such as PACS, Office-31 and Office-Home. The results in the Table 1 (see the attached pdf) show that our method outperforms the compared methods in these settings with a large margin, which confirms the advantage of our method.
>
> 2. **Can the combination of intra-class compactness and Dispersion Score performs better than only Dispersion Score?**
>
> Thank you for the suggestion. In the Table 3 (see the attacked pdf), we show that combining dispersion with compactness cannot outperform using Dispersion Score only.
>
> [1] Shen, Z., Liu, J., He, Y., Zhang, X., Xu, R., Yu, H., & Cui, P. (2021). Towards Out-Of-Distribution Generalization: A Survey. ArXiv, abs/2108.13624.

---

> > ### Comment · Reviewer_ALm9 · 2023-08-18
> > **Thanks for the response!**
> >
> > Thanks for the response!
> >
> > Most of my questions have been solved. I would like to increase the score by one level.
> >
> > I sincerely suggest the authors include a discussion about the difference with OOD detection and add some works of literature. This would help readers to better understand the work.

---

> > > ### Author Response · Authors · 2023-08-20
> > >
> > > Thank you for checking our rebuttal and raising your score. We will add the discussion in the related work as your suggested. Sincerely thanks for your valuable time on this paper!

---

### Official Review · Reviewer_BweV · 2023-07-03

**Soundness:** 4 excellent
**Presentation:** 3 good
**Contribution:** 4 excellent
**Rating:** 8
**Confidence:** 5

**Summary:**

In this paper, the authors focus on the task of predicting model performance on unseen/shifted datasets, without support of annotations. To do so, they first show the connection between feature separability and test accuracy, with an intuitive example and theoretical explanation. Based on the analysis, they propose a novel metric, Dispersion score, which measures the inter-class divergence from feature representations. They also reveal that intra-class compactness does not reflect the generalization performance, while inter-class dispersion works as a good indicator. They conduct experiments on CIFAR-C and TinyImageNet-C datasets to show the efficiency and effectiveness of the proposed metric. Furthermore, they also show the advantages of their method in some extreme cases (limited data, partial label set, class imbalance).

**Strengths:**

1.The motivation of this work is reasonable. Intuitively, the prediction performance should be tightly tied with the quality of the learned feature. High inter-class dispersion is one of the goals of self-supervised learning for learning a good representation. The authors also provide an interesting explanation from a theoretical perspective, which is one of the highlights of this work.

2.The analysis of those methods with distribution distance is thought-provoking. In my view, the shifted distance might not be necessarily connected to the test performance, as the shifted features might not be important for the final prediction. For example, an image classifier with sufficient generalization ability would not change its predictions when the background is changed a lot.

3.The proposed method is novel and interesting. To the best of my knowledge, this is the first work to exploit the feature properties of test instances for predicting OOD error. It does not require access to the training data and only uses a forward propagation, which is much faster than existing SOTA methods.

4.The empirical results are extensive and convincing. The authors not only show the simple method outperforms existing training-free methods, but also compare to ProjNorm on the computational efficiency (575 second v.s. 11 second). The most exciting part for me is the analysis on the flexibility in OOD data, which considers some real-world settings, like class imbalance and limited data. The authors also present that a high intra-class compactness is not necessary for good prediction performance, which may provide a new insight for representation learning.

**Weaknesses:**

1.The presentation of some Figures can be improved. For example, in Figure 1b, the magnified box seems not match the original area. The four images of Figure 2 are a little small, which might be improved by reducing the blanks between images.

2.Why intra-class compactness does not work is not clear. Although the authors show that intra-class compactness is not an effective indicator empirically, it could be better if the authors can provide an intuitive explanation.


**Questions:**

see weaknesses.

**Limitations:**

The authors discussed the limitations of the proposed score in the adversarial setting, where the feature quality is also broken by adversarial attacks. The authors also provide analysis to show the underlying reason, which is also an interesting contribution.

---

> ### Author Rebuttal · Authors · 2023-08-09
>
> 1. **Improving quality of some figures.**
>
> Thanks for your constructive suggestions. We will improve the quality of those figures in the revised version, as you suggested.
>
> 2. **The reason why intra-class compactness dose not work.**
>
> Thank you for the thought-provoking question. Previous works in representation learning demonstrate that high intra-class compactness and inter-class separability are correlated to the final accuracy in machine learning. In this paper, we show that intra-class compactness cannot be used to indicate the prediction performance under distribution shift, while inter-class separability still works. The potential reason of the difference is from the effects of convariate shifts. The convariate shift may have a significant influence on the intra-class compactness so that compactness cannot effectively reflect the final accuracy. We will explore this hypothesis in our future work.

---

> > ### Comment · Reviewer_BweV · 2023-08-11
> > **To response**
> >
> > Thank you for the detailed response. My concerns have been addressed. I also read the other reviews and your responses. You have done a good job and I believe those discussion can further enhance this work.

---

> > > ### Author Response · Authors · 2023-08-20
> > >
> > > Thank you for reading our response and keeping a positive score! We are really grateful for your time and expertise.

---

### Official Review · Reviewer_HxyT · 2023-07-06

**Soundness:** 4 excellent
**Presentation:** 4 excellent
**Contribution:** 3 good
**Rating:** 7
**Confidence:** 4

**Summary:**

This research focuses on predicting test accuracy on shifted datasets without access to ground-truth labels. The authors begin by analyzing the potential issue of the existing methods which were based on the shift distance and point out that the previous distributional distances were not always correlated highly to the out-of-distribution (OOD) error. Then, they proceed with an intuitive example and theoretical explanation, showing the connection between feature separability and test accuracy. Based on this, they propose a novel metric that measures the inter-class dispersion, which is demonstrated as an effective factor for the OOD error estimation. They conducted experiments on CIFAR-C and TinyImageNet-C to validate the advantages of the proposed method. The authors further demonstrate the robustness of the proposed method against class imbalance and data shortage.

**Strengths:**

1. The task studied in this paper is practically important. In some real-world applications, it is necessary to assess the model performance on a given unlabelled dataset. Under those scenarios, OOD error estimation becomes inevitable and valuable.
2. The proposed method is supported by both empirical observations and theoretical analysis. The motivation is clear.
3. The proposed method is novel, effective, and efficient. Previous SOTA methods like ProjNorm need to update the model which is computationally expensive (It can be unachievable with some large models). They also show that intra-class compactness cannot work well, which motivates deeper exploitation of the properties within the features distribution rather than the coarse-grained property on the whole dataset adopted in AutoEval.
4. It is also interesting to see that this paper further investigated the performance under some scenarios with imperfect data, e.g., class imbalance, smaller sample size, and partial OOD error prediction. To the best of my knowledge, this is the first work providing such a complete analysis. Compared with the previous studies, the proposed approach achieved comparable performance even under these extreme cases.
5. This paper is well-written and easy-to-understand. The analysis and figures provided by the authors are clear and informative. I believe readers can easily get the core idea and implement it.

**Weaknesses:**

1. Some potential typos should be corrected in the next version. See the Questions part for more details.
2. The discussion about the pseudo labels used for cluster centroid determination can be further extended. See Questions for more details.

**Questions:**

- I noticed that when computing the dispersion score, the pseudo labels were used to determine the cluster centroid. In Appendix E, the author compared the pseudo-labeling with another clustering method, K-means. However, I am interested in if it can be further improved by improving the quality of these pseudo labels. For example, filtering out some low-confidence predictions, adopting soft pseudo labels instead of hard ones, or considering wrong pseudo labels as noisy annotations?
- I also noticed that the previous work, ProjNorm, also adopted pseudo labels for the OOD error prediction. Could you please summarize the differences in how to adopt the pseudo labels between the two methods?
- About the evaluation times of Dispersion score on TinyImageNet (Table 2). The time of ResNet50 should be longer than that of ResNet18. Is there a typo that records the times in the wrong order?
- From the analysis in 4.4, does it mean that we can focus more on the inter-class dispersion instead of intra-class compactness in representation learning?
- Some potential typos:
    - Line 221, Spearson -> Pearson?
    - Line 240, Table 6 -> Table 1?

**Limitations:**

I did not see any severe limitations in this paper.

---

> ### Author Rebuttal · Authors · 2023-08-09
>
> 1. **Can improving the quality of pseudo labels further enhance performance of OOD error estimation?**
>
> To verify whether improving quality of pseudo labels can further improve the estimation performance, we conduct experiments on Tiny-ImageNet with ResNet18 by filtering out low-confidence predictions under a certain threshold. From the results below, we can observe that selecting high-confidence samples even degrades the estimation performance. Furthermore, we present in Appendix D that using ground-truth labels cannot improve the estimation performance, showing that improving label quality may not be a correct direction. As we discussed in the response to the reviewer 1A8L, pseudo labels in our method is used to introduce the bias of the linear layer, so a potential direction might be using the softmax output, instead of the one-hot pseudo labels.
>
>
> |Threshold|0.0|0.05|0.1|0.2|0.3|0.4|0.5|0.6|0.7|0.8|0.9|
> | --- | - | - | - | - | - | - | - | - | - | - | - |
> ||0.966|0.965|0.966|0.945|0.917|0.886|0.875|0.847|0.821|0.798|0.786|
>
>
>
> 2. **The difference between Dispersion and ProjNorm in adopting pseudo labels.**
>
> Thank you for the great question. Yes, both our Dispersion Score and ProjNorm adopt pseudo labels during OOD error estimation, but we note that the role of the pseudo labels in these two methods are somewhat different. In ProjNorm, they use pseudo labels in a "self-training" manner, which is commonly used in semi-supervised learning. As the number of mislabeled out-of-distribution (OOD) examples increases, the training deviation of parameters tends to expand further. In this way, the distance in parameter space can be used to measure the test accuracy. In our method, pseudo labels is used to introduce the bias of the linear layer, which also affects the final predictions. If we simply use the true labels, Dispersion score can only measure the quality of the learned representations, instead of the final accuracy.
>
> 3.  **Do we need to pay more attention on inter-class dispersion than intra-class compactness in representation learning?**
>
> In this work, our findings indicate that there is no significant linear correlation between intra-class compactness and test accuracy on out-of-distribution datasets. However, there might be some other forms of relationship between them, e.g., nonlinear correlation. Therefore, it may needs more analyses to explore the specific effects of intra-class compactness in representation learning， which we believe can benefit from the insight from this work.
>
>
> 4. **Some typos**
>
> Many thanks for your suggestions. We will fix those typos in the revised version.

---

> > ### Comment · Reviewer_HxyT · 2023-08-12
> > **Response to Author Rebuttal**
> >
> > Thanks for the response from the authors. My concerns have been addressed. In addition, I attached some further comments w.r.t. the response from the authors as follows:
> >
> > Firstly, it is great to see that the authors provided additional experiments by adopting different confidence levels w.r.t. the pseudo label. The results indicated that filtering out samples with a confidence threshold is even detrimental to the final performance. According to my understanding, the model obtained on the training set can be poorly calibrated on the OOD dataset (usually we do not apply a calibration process in OOD error prediction), and discarding low-confident samples with a threshold can be considered as introducing more over-confidence in disguise during calculating the dispersion score. Is this a reasonable explanation?
> >
> > Secondly, it is also interesting to see the results on the open-set OOD prediction setting (Table. 2 of pdf attachment). Although the current OOD error prediction task does not consider the existence of label space change, it is still interesting to see that the proposed method outperforms the baselines under the constructed open-set setting.

---

> > ### Author Response · Authors · 2023-08-12
> >
> > Dear Reviewer HxyT,
> >
> > Thank you so much for recognizing this work. In our understanding, filtering out test samples by a confidence threshold has two negative impacts on OOD error estimation: information loss and high bias. Firstly, the true accuracy is calculated by two parts of information: correctly-classified samples and wrongly-classified samples. If we filter out those samples with low confidence, the calculated metric may ignore information from wrongly-classified samples, leading to the degradation of the estimation performance. Secondly, the number of test samples for calculating the metric is reduced after filtering out samples with lower confidence. Therefore, the calculated metric would be sensitive to those remaining samples that are biased toward high confidence (corresponding to high accuracy). With these two effects, the metric would tend to give overly optimistic estimations if we only use examples with high confidence.
> >
> > Best Regards,
> >
> > Authors

---

> > > ### Comment · Reviewer_HxyT · 2023-08-18
> > > **From the reviewer**
> > >
> > > Thank you for addressing my questions. Based on the answers and discussions in other threads, I will keep my score as 7 and continue to support the acceptance of this work.

---

> > > > ### Author Response · Authors · 2023-08-20
> > > >
> > > > Thank you for reading our response and supporting this work! We are really grateful for your time and expertise.

---

### Official Review · Reviewer_1A8L · 2023-07-06

**Soundness:** 2 fair
**Presentation:** 2 fair
**Contribution:** 2 fair
**Rating:** 5
**Confidence:** 5

**Summary:**

This work aims to predict classifier accuracy on unlabeled test samples. To achieve this goal, this paper proposes a feature separability-based dataset-level score to check whether features have high inter-class scatter. This feature separability score is calculated by measuring how far the centroids of features that share the same pseudo-label predicted by the model deviate from the center of all features on average. The experiments show that such inter-class dispersion is strongly correlated with model accuracy.

**Strengths:**

+ [***Good clarity***] This work is well-written and easy to follow. The method is well presented, the visualizations are helpful, and the experimental settings are clearly introduced.

+ [***Measuring feature separability is well-motived***] Under distribution shifts, the features of the source and target can be scattered differently. Using such information to reflect model accuracy seems reasonable.

**Weaknesses:**

- [***More results to illustrate the relationship between distribution gap and accuracy***] In the accuracy prediction, there are two metrics are proposed to measure the distribution gap. Please show their results in Figure 1 to well illustrate the motivation regarding the potential limitation of the distribution gap for accuracy estimation.

- [***The definition of dispersion score is not sound***] The class cluster is based on the classifier's prediction. What if the classifier gives bias predictions on test sets? For example, in adversarial examples, the classifier maintains class-class separability but totally misclassified data. Moreover, why using the gound-truth label to define class clusters does not give stronger correlation strength (Section D).

    Moreover, whether the proposed method can handle the cases where some classes do not appear or some unseen classes appear. For example, ProjNorm discusses the label shift where some classes are missing. Under such a scenario, ProjNorm is less effective than other methods.

- [***The experimental setting is somewhat limited***] This work only provides the experiments on small-scale datasets (e.g., CIFAR-10/100 and TinyImageNet). Considering the literature, the results on iWILDS and ImageNet should be included. For example, DoC and ATC report the results on ImageNet datasets with several natural distributions, such as ImageNet-V2, ImageNet-R and ObjectNet. Without the results on such realistic datasets, it is hard to conclude the robustness and effectiveness of the proposed method.

**Questions:**

- Why using the gound-truth label to define class clusters does not give stronger correlation strength (Section D).
- Under an adversarial attack, the features might still have high dispersion but the classifier archives low accuracy. Please comment on this and discuss the potential solution to alleviate this.
- Class imbalance results (Table 3) are not sufficient. How about other existing methods (e.g., DoC and ATC) under such class imbalance?

**Limitations:**

This work reports results for some special cases, such as class imbalance and adversarial attacks. Another potential limitation is the open set problem, where some unseen classes arise. Also, some classes may be missing during testing. It would be better to mention and discuss both cases, as the proposed scatter score can be significantly affected.

***[Post-rebuttal]***

> I would recommend excluding the open-set results from the main paper due to the evaluation metric's limitation in assessing only the seen classes. Moreover, ATC reports the results on some real-world and large-scale datasets like iWILDS and ImageNet, including such datasets would make the submission solid. This addition would enhance the robustness of your research and highlight its practical applicability. Given that you've already showcased results on domain adaptation datasets such as PACS, Office-31, and Office-Home, I am inclined to think that the current evaluation is sufficient.

> [Additional suggestions which do not impact the rating of this paper] You might consider referring to two relevant works: "Unsupervised Accuracy Estimation of Deep Visual Models using Domain-Adaptive Adversarial Perturbation without Source Samples," which also features results on domain adaptation datasets, and "Characterizing Out-of-Distribution Error via Optimal Transport." This latter work, which assumes a consistent marginal label distribution between training and test sets, stands in contrast to your method and may help highlight the merits of your approach.

---

> ### Author Rebuttal · Authors · 2023-08-09
>
> Great thanks for your constructive comments and suggestions! Please find our response below.
>
> 1. **Improving Figure 1 by adding the two matrics.**
>
> Thank you for the suggestion. In the final version, we will add the corresponding metrics (shown in table 1) on the figures. We list the numerical details below for your reference.
>
> |  $Method$ | $Fr\acute{e}chet$ | $MMD$ |
> | ---- | ---- | --- |
> | $R^2$ | 0.858 | 0.804 |
> | $\rho$ | 0.964 | 0.943 |
>
>
> 2. **Can Dispersion Score work under adversarial attacks?**
>
> In this paper, our method is designed for predicting the test accuracy on OOD data, which is generally produced by natural corruption or perturbation. As an extension, we discussed the adversarial setting in Appendix H. The results show that those methods of OOD error estimation cannot provide meaningful performance under adversarial attacks, as well as our method. This is consistent with the observation described in ProjNorm, where their method predicts an error of 28.1% when the true error is 100.0%.
>
> Furthermore, we find that feature dispersion may provide an intuitive understanding of the inherent difference between adversarial and (natural) corruption robustness. As shown in Figure 7, adversarial perturbations surprisingly increase the distance between different clusters, while corruption perturbations decrease the separability of the clusters. The results reveal that adversarial perturbations affect the model predictions in a different way: assigning instances to the wrong groups and enlarging the distance among those groups. This explains why dispersion score cannot be applied to predict the performance on adversarial examples. While the limitation on adversarial examples persists (like ProjNorm), we hope the above insight can inspire specific designed methods for predicting adversarial errors in the future.
>
> 3. **The performance of Dispersion Score in parial OOD error prediction.**
>
> In Appendix G, we provide an analysis for the parial setting, where the label space of the test dataset is only a subset of that of the training dataset. In Table 7, we can observe that our method achives better performance than existing methods when some classes disappear during testing.
>
> 4. **The performance of Dispersion Score when the test set includes unseen classes.**
>
> Many thanks for your constructive suggestions. We conduct this experiment by injecting 10% extra examples from unseen classes with pretrained ResNet18. We use Tiny-ImageNet and CIFAR-10 as the ID datasets, while 300K Random_Images and CIFAR100 as the datasets with unseen classes. The results are presented in Table 2 (see the attached pdf), which shows that our method outperforms previous methods.
>
> 5. **Why using the true labels cannot outperform our method with pesudo labels?**
>
> It is because Dispersion score with ground-truth labels only measures the quality of the learned features, regardless of the last linear layer. In an extreme case with a perfect feature extractor, the accuracy might be low due to the poor linear classifier, even though Dispersion score with ground-truth labels is very high. Using pesudo labels, Dispersion score involves in the bias of the linear classifier, thereby showing better performance in predicting OOD errors. We may provide a formal analysis to show the advantages of pesuado labels in Dispersion score in the final version or future work.
>
> 6. **More results on datasets with several natural distributions.**
>
> To illustrate the effectiveness of Dispersion Score under realistic datasets, we evaluate the estimation performance via MSE on ImageNet-R with WRN-50-2. Their results are shown below.
>
> |MSE|Entropy|ATC|Frechet|ProjNorm|Dispersion|
> | - |- |- |- |- |- |
> |ImageNet-R|27.67|13.27|11.31|7.23|$\textbf{4.46}$|
>
> In addition, we also evaluate our method under datasets with more complex distribution shift, such as PACS, Office-31 and Office-Home, which results are shown in Table 1 (see the attached pdf). We can observe from those tables that Dispersion Score obtain better and more stable performance compared with other compared baselines, while the state-of-the-art method, ProjNorm, almost fail in these cases.
>
> 7. **The results of other compared methods in the imbalanced setting.**
>
> We show the results of other compared methods in Appendix F. In Table 6, we illustrate the performance of the rest methods including Rotation, Entropy, ConfScore, AgreeScore, ATC and $Fr\acute{e}chet$ under the imbalanced setting. The results show that our method outperforms those compared methods by a meaningful margin.

---

> > ### Comment · Reviewer_1A8L · 2023-08-12
> > **Follow-Up Discussion**
> >
> > Dear Authors,
> >
> > Thanks for the response!
> >
> > - I am interested in the provided "The performance of Dispersion Score when the test set includes unseen classes". With new classes, how to evaluate classifiers? Viewing the unseen classes as one outliner class or just evaluating on original seen classes?
> >
> > - "Why using the true labels cannot outperform our method with pesudo labels?" This part is still not clear to me, especially "Dispersion score with ground-truth labels only measures the quality of the learned features, regardless of the last linear layer".
> >
> > Best,
> > Reviewer 1A8L

---

> > > ### Author Response · Authors · 2023-08-12
> > >
> > > Dear Reviewer 1A8L,
> > >
> > > Thank you for the further questions and we are glad to have an in-depth discussion with reviewers. Please find our response below.
> > >
> > > 1. **How to evaluate the model performance when the test set includes examples from unseen classes?**
> > >
> > > For this setting, we only calculate the accuracy on the original seen classes. Despite that we do not care about the performance on those examples from unseen classes in this task, those open-set samples may have a large impact on the metrics, which are used to estimate the prediction performance (as mentioned by Reviewers 1A8L and ALm9). Our results show that Dispersion score can achieve SOTA performance in this setting. Please kindly let us know if there are any recommended evaluation methods for this setting.
> > >
> > > 2. **Why Dispersion score with ground-truth labels only measures the feature quality?**
> > >
> > > We would like to clarify that the final predictions of a deep network depend on both the learned feature $\boldsymbol{z}$ and the linear classifier $f_{\omega}$: $\boldsymbol{p}=f_{\omega}(\boldsymbol{z})$. If we calculate the Dispersion score with the ground-truth labels, the metric will be not related to the linear classifier $f_{\omega}$, as it only uses the feature extractor $f_g$ and the ground-truth labels. In this manner, Dispersion score with the ground-truth labels does not contain the biased information of the learned classifier, thereby being suboptimal in estimating the final prediction performance. We also give an extreme case in the last response, which may provide an intuitive understanding of this part.
> > >
> > > Best regards,
> > >
> > > Authors

---

> > > ### Author Response · Authors · 2023-08-20
> > >
> > > Sincerely thanks for your informative feedback and valuable time in the discussion. We will incorporate the new results and explanations into the final version appropriately. We are open to further discussion if there are any remaining concerns.

---

### Author Rebuttal · Authors · 2023-08-09

## General Response

We thank all the reviewers for their time, patient, and valuable comments. We are glad that all reviewers agree that our work is *well-motivated and well-organized, clear, and concise*. Reviewers HxyT, BweV, and ALm9 appreciate that *the experimental results are strong and convincing*. We are also encouraged that reviewers find this method is *novel, comprehensively analyzed (BweV, HxyT), and easy-to-use (Alm9)*.

We respond to each reviewer's comments in details, respectively. And we put three tables in the attached pdf file, which support our claims in the responses. In the revised version, we will also update the manustript according to reviewers' suggestions, and we believe this makes our paper much stronger.

---

### Decision · Program_Chairs · 2023-09-21

**Decision:**

Accept (poster)

**Comment:**

All reviewers voted to accept the paper, citing the importance of the problem being tackled, the empirical and analytical analysis, and clarity of writing.